# Opening gap width influences distal tibial rotation below the osteotomy site following open wedge high tibial osteotomy

**Jun-Ho Kim, Hoon-Young Kim, Dae-Hee Lee** *

Department of Orthopaedic Surgery, Samsung Medical Center, Sungkyunkwan University School of Medicine, Seoul, Korea

* eoak22@empal.com

## Abstract

### Purpose

Although rotation of the distal portion of the tibia below the osteotomy site is considered an inevitable change in the axial plane in open wedge high tibial osteotomy (HTO), several studies on this issue have shown contradictory results. The purpose of this study was, therefore, to determine the direction and amount of distal tibial rotation following open wedge HTO using a three-dimensional reconstructed model.

### Methods

This study involved 41 patients (42 knees) undergoing open wedge HTO for primary medial osteoarthritis. Distal tibial rotation was measured on the overlaid tibial plateau of a preoperative and postoperative 3-dimensional reconstructed model based on computed tomography.

### Results

The mean distal tibial external rotation was 2.7° ± 2.3° (range, -0.9° to 9.9°), and the opening gap was larger in the group with > 3° distal tibial rotation than in the group with ≤ 3° distal tibial rotation (11.4 mm vs. 9.6 mm, P = 0.027). Multiple regression analysis showed that the opening gap was the only predictor of distal tibial rotation. On receiver operating characteristics analysis, an opening gap of 10 mm was found to be the optimal cutoff value for achieving greater than 3° of distal tibial rotation.

### Conclusions

Following medial opening wedge HTO, the distal tibial portion below the osteotomy site rotated approximately 3° externally. The magnitude of the external rotation of the distal tibia was affected by the opening gap width.

**Data Availability Statement:** All relevant data are within the paper its supporting information file.

**Funding:** The authors received no specific funding for this work.

**Competing interests:** The authors have declared that no competing interests exist.

## Introduction

Open wedge high tibial osteotomy (HTO) is an established procedure that provides pain relief and functional improvement in patients with medial knee osteoarthritis and varus deformities. [1–6] The biomechanical rationale of this procedure is to shift joint loading from the medial compartment, which has osteoarthritic cartilage, to the lateral compartment with relatively intact cartilage in the coronal plane.[2] The alignment change of the coronal plane could lead to unintended consequences in the sagittal and axial planes because of the three-dimensional (3-D) characteristics of the proximal tibia.[7–9] The change of the posterior tibial slope in the sagittal plane, which is one of the unintended consequences of open wedge HTO, is now well known to orthopedic surgeons, and could be used as another treatment modality for knee instability by intentional control of the posterior tibial slope change.[10] Unlike the posterior tibial slope change, the rotation of the distal portion of the tibia under the osteotomy gap has not been well studied[11] although it also inevitably changes in the axial plane in open wedge HTO. In addition, previous studies[12–14] dealt with distal tibial fragment rotation and used various unreliable measurement methods, such as intraoperative visualization of landmarks, Kirschner wire, or the axial plane of two-dimensional computed tomography (CT), and thus, contradictory results were obtained regarding the direction and the amount of distal tibial rotation. It is unclear whether the distal part under the osteotomy site rotates internally or externally, and if it does rotate, what the degree of rotation is also remains unclear.

The purpose of this study was, therefore, to determine the direction of rotation (whether internal or external) and the amount of rotation in the distal part of the tibia below the osteotomy gap following open wedge HTO using a 3-D reconstructed model. We hypothesized that the distal tibia below the osteotomy site would not rotate significantly following medial opening wedge HTO.

## Materials and methods

### Patients

The institutional review board of Samsung Medical Center (IRB No. 2017-02-030-001) approved this retrospective study and waived the requirement for informed patient consent. A retrospective review was conducted of 41 consecutive patients (42 knees) who underwent biplanar medial open wedge HTO for medial osteoarthritis with varus deformities at our institution. Patients were considered ineligible for HTO if they had symptomatic osteoarthritis of the patellofemoral joint and lateral compartment, rheumatoid arthritis, a knee range of motion $< 100°$, grade $\geq 3$ lateral collateral ligament laxity, or a flexion contracture $> 10°$. All patients agreed to undergo CT scanning before and after surgery. The patients' demographic information is summarized in Table 1 (S1 File).

### Surgical technique

All surgical procedures were performed by a senior surgeon. The preoperative amount of correction and the size of the osteotomy gap were assessed on standing lower limb radiography. An anteromedial vertical skin incision was made from the superomedial aspect of the inferior patellar pole to 4 to 5 cm below the tibial tubercle. The patellar tendon was exposed for vertical osteotomy. After identifying the pes anserine tendon, the fascia overlying tendon was incised and release toward posterior capsule and pes anserine tendon was detached from the tibia. Also, the superficial medial collateral ligament (MCL) was identified and distal superficial MCL was carefully released. Then, posterior capsule was incised vertically along the posterior margin of tibia and was released from the tibia. A guidewire was inserted with visual assistance

**Table 1. Summary of preoperative patient characteristics.**

| Parameter | |
|---|---|
| Number of patients (male : female) | 9 : 32 |
| Age, years* | 56.7 (44 to 73) |
| Height, cm* | 159 (145 to 180) |
| Weight, kg* | 70 (49.8 to 95.6) |
| Body mass index, kg/m²* | 27.5 (21.6 to 33.4) |
| Deformity, degrees† | Varus 10.4 ± 3.4 (1 to 15) |
| Preoperative range of motion, degrees† | 120.4 ± 11.6 (105 to 140) |

*Median (range)

†Mean ± standard deviation (range)

using an image intensifier on fluoroscopy, and the osteotomy was performed with a saw, under the guidance of an image intensifier, for a distance of up to 1 cm from the lateral cortex. The oblique osteotomy part of the biplane osteotomy was intended to commence at the medial tibial cortex along the metaphyseal flare (approximately 3 to 4 cm distal to the joint line) and was angled in such a way that it terminated at the level of the tip of the fibular head laterally. Efforts were made to keep the lateral cortex and lateral capsular hinge intact. The vertical osteotomy part of the biplane osteotomy was performed at the posterior aspect of the tibial tubercle, thus avoiding violation of the bony attachment of the patellar tendon. Following this, the bone at the site of the osteotomy was forced to open carefully and gradually, using a double chisel, to its target width. When the mechanical axis passed through 62% of the tibial plateau, as assessed by intraoperative fluoroscopy, and the target osteotomy width was achieved, the osteotomy was stabilized using a fixed-angle plate with interlocking screws (TomoFix™, Synthes, Bettlach, Switzerland). The gap on the medial side was filled with a cancellous bone chip allograft.

## Plain radiography

For the preoperative and postoperative radiographic evaluation, an anteroposterior (AP) lower limb view on standing radiographs was obtained. Using Centricity RA 1000 (GE Medical System, Milwaukee, Wisconsin), a type of picture archiving and communication system, the mechanical femorotibial angle (mFTA) was measured. The mFTA was defined as the angle of two lines; one was the femoral axis between the center of the hip and the center of the notch in the distal femur, and the other was the tibial axis between the center of the tibial spines and the center of the ankle. A postoperative lower limb standing radiograph was performed 12 weeks after surgery when the patient was able to stand without any support.

## CT analysis

All CT data were obtained by the same CT scanner (Lightspeed VCT; GE Medical Systems, Milwaukee, WI). The collimation was 16 x 0.625 mm, the slice thickness was 0.625 mm, and the acquisition matrix was 512 x 512. Postoperative CT scan was performed on postoperative day 5 after removing the hemovac drain. Lateral hinge fracture classified by Takeuchi's method was detected by thorough investigation of coronal images. The vertical osteotomy direction was determined by the angle between vertical osteotomy line and posterior margin of tibia plateau on axial images (Fig 1A and 1B). The angle of biplane osteotomy was evaluated between vertical and oblique osteotomy at the level of tibial tuberosity on sagittal images (Fig 1C).

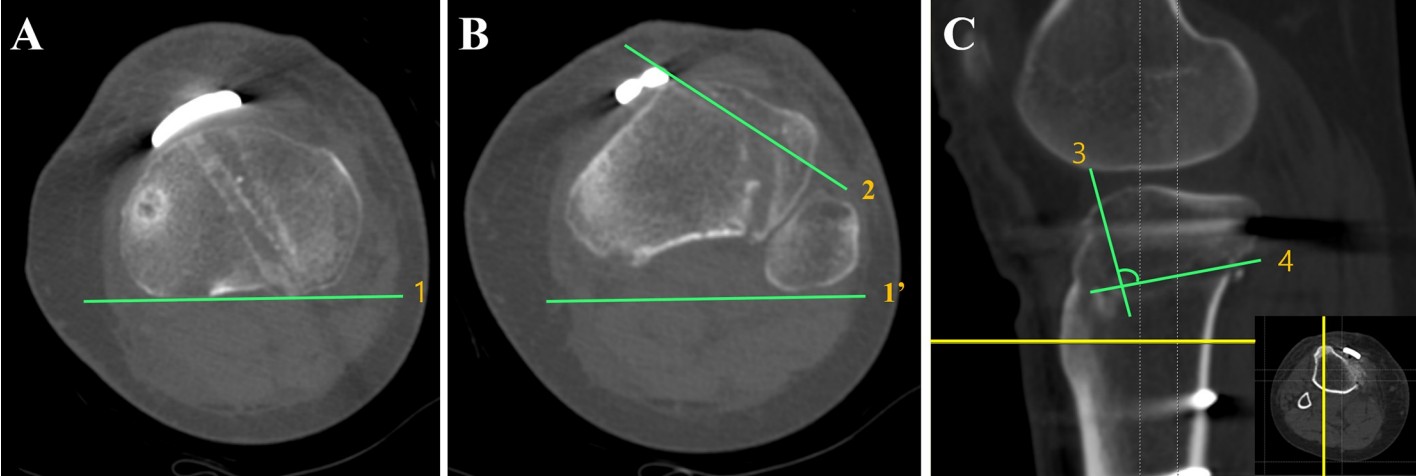

**Fig 1. Measurement of the angle of vertical osteotomy direction and biplane osteotomy angle on computed tomography. (A)** Line 1, a tangent line to posterior tibia plateau is drawn on the axial image of tibia plateau level. **(B)** Line 1', copied and pasted from Fig 1A. The angle of vertical osteotomy direction is determined between line 1' and vertical osteotomy (line 2). **(C)** A sagittal image at the prominent tibial tuberosity level. The biplane osteotomy angle is determined between vertical osteotomy (line 3) and oblique osteotomy (line 4).

### 3-D-CT data acquisition and 3-D model data analysis

After DICOM (Digital Imaging and Communication in Medicine) data were extracted by a picture archiving and communication system, they were exported to Mimic (Materialise, Leuven, Belgium), and the 3-D model of the proximal tibia was created.

The 3-D models of the proximal tibia were exported into Geomagic 2013 (Geomagic, Research Triangle Park, NC, USA). True axial, AP, and lateral views were created by manipulating the 3-D model in Geomagic. True axial and AP views were created by the following processes. The AP axis was determined by connecting the center of the tibial insertion of the posterior cruciate ligament to the medial border of the tibial insertion of the patellar tendon, as described by Akagi et al.[15] The flat surface of the medial tibial plateau was selected, and then we made the best-fit plane to the surface. From a bird's eye view, the model of the proximal tibia was rotated axially until the AP axis of the model was aligned on the AP axis of the 3-D space in the Geomagic program, which is the true axial view. The model was then rotated sagittally until the best-fit plane was aligned as a line on the 2-D projected plane, which is the true AP view. After obtaining the true AP view, the model was axially rotated 90˚, which is the true lateral view.

To evaluate the postoperative change of the axial rotation of the distal tibia, we overlaid the tibial plateau of the preoperative and postoperative 3-D model and evaluated the change of the tibial tubercles because the tibial tubercle is distal to the osteotomy site. The identical position was marked on the tibial tubercle in the true AP view on both the preoperative and postoperative 3-D models (Fig 2A and 2B), and the marking was checked ensure it was the most prominent apex of the tibial tubercle in the true axial and lateral views. The overlay model of the preoperative and postoperative tibial plateau can be achieved using the Geomagic program, considering the definite landmarks on the tibial plateau as follows: (1) anterior border of the medial plateau; (2) posterior border of the medial plateau; (3) medial border of the medial plateau; (4) anterior border of the lateral plateau; (5) posterior border of the lateral plateau; (6) lateral border of the lateral plateau; and (7)-(9) intercondylar eminences.

To measure the angle of the axial rotational change of the distal tibia after HTO, two lines were drawn from the sulcus of the posterior border of the tibial plateau to the marked

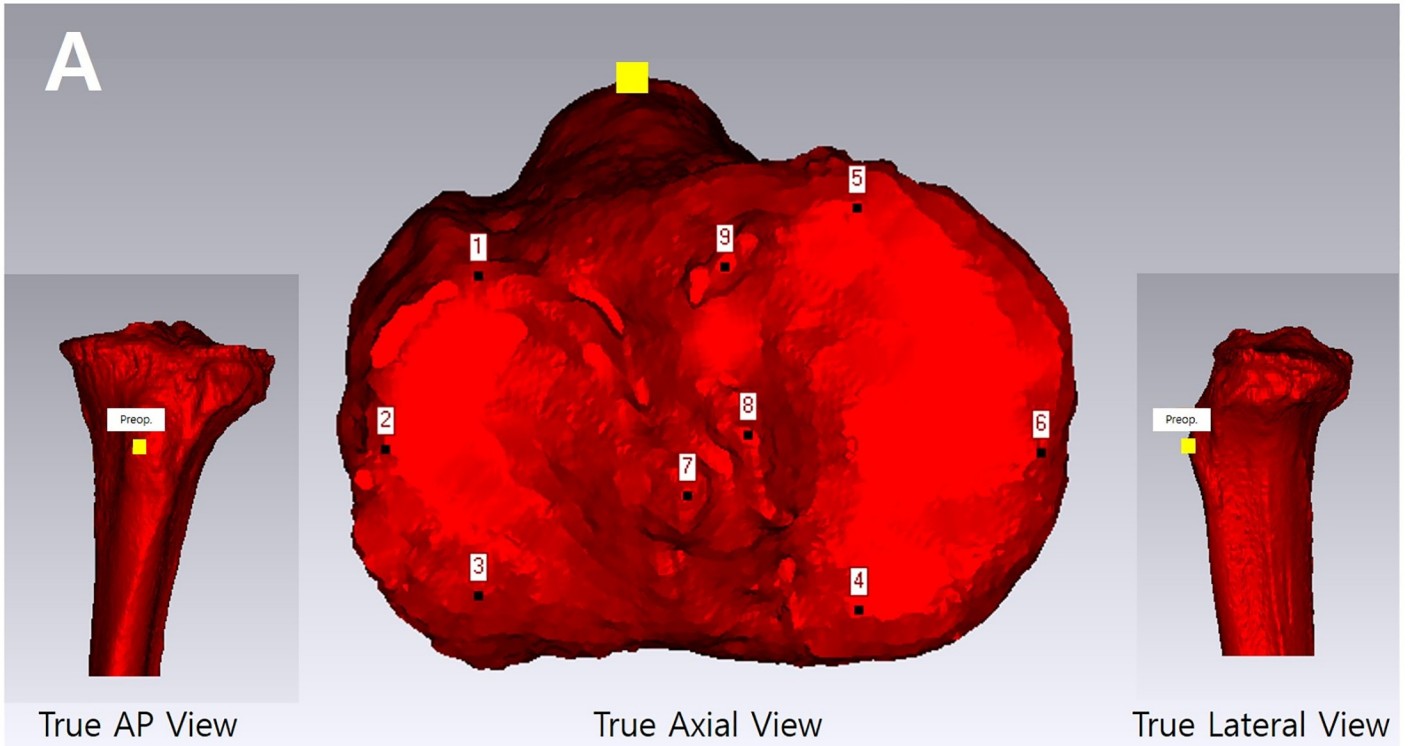

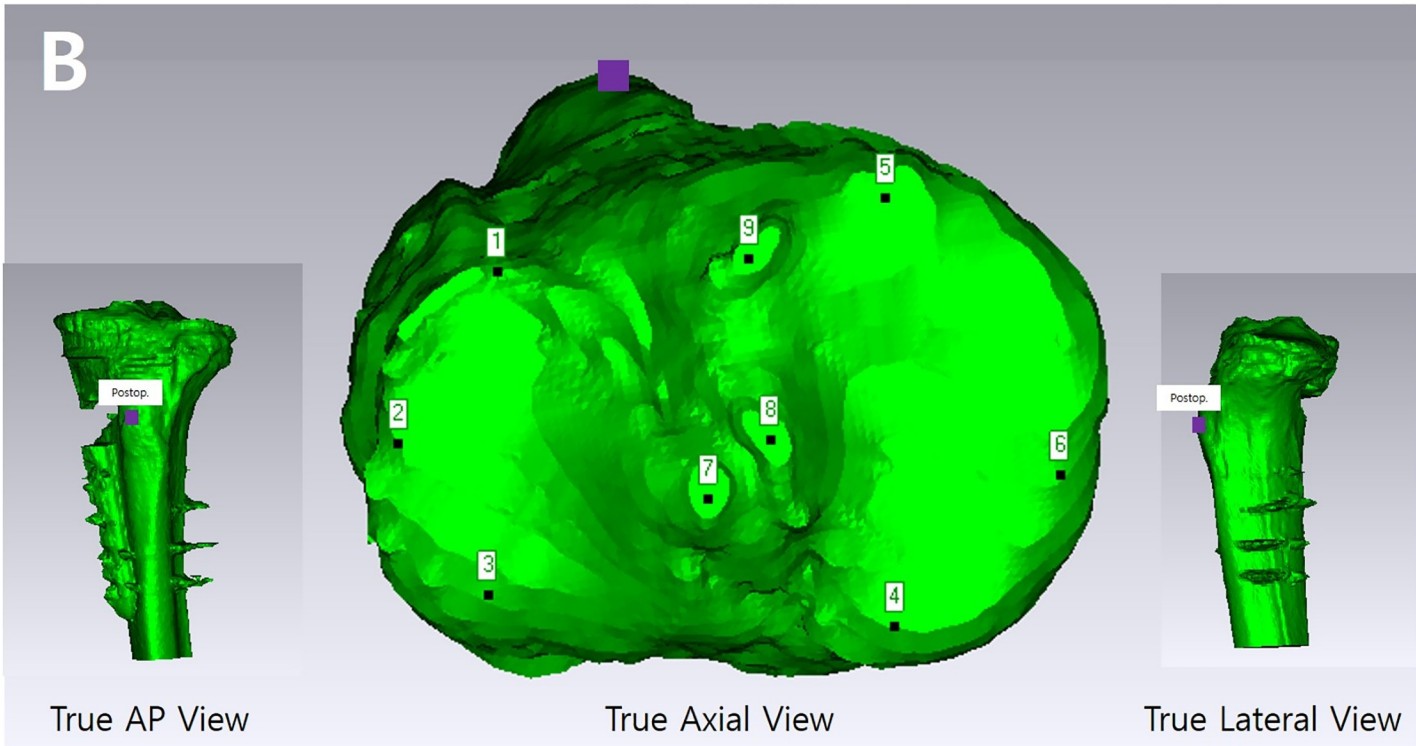

**Fig 2. Three-dimensional (3-D) reconstructed model of the proximal tibia based on preoperative and postoperative computed tomography images.** On the axial plane of the preoperative (A) and postoperative (B) 3-D reconstructed model, the tibial tuberosity positions (yellow and purple squares on preoperative and postoperative 3-D models, respectively) were determined, which correspond to the same point of the tibial tuberosity on both the coronal and sagittal planes.

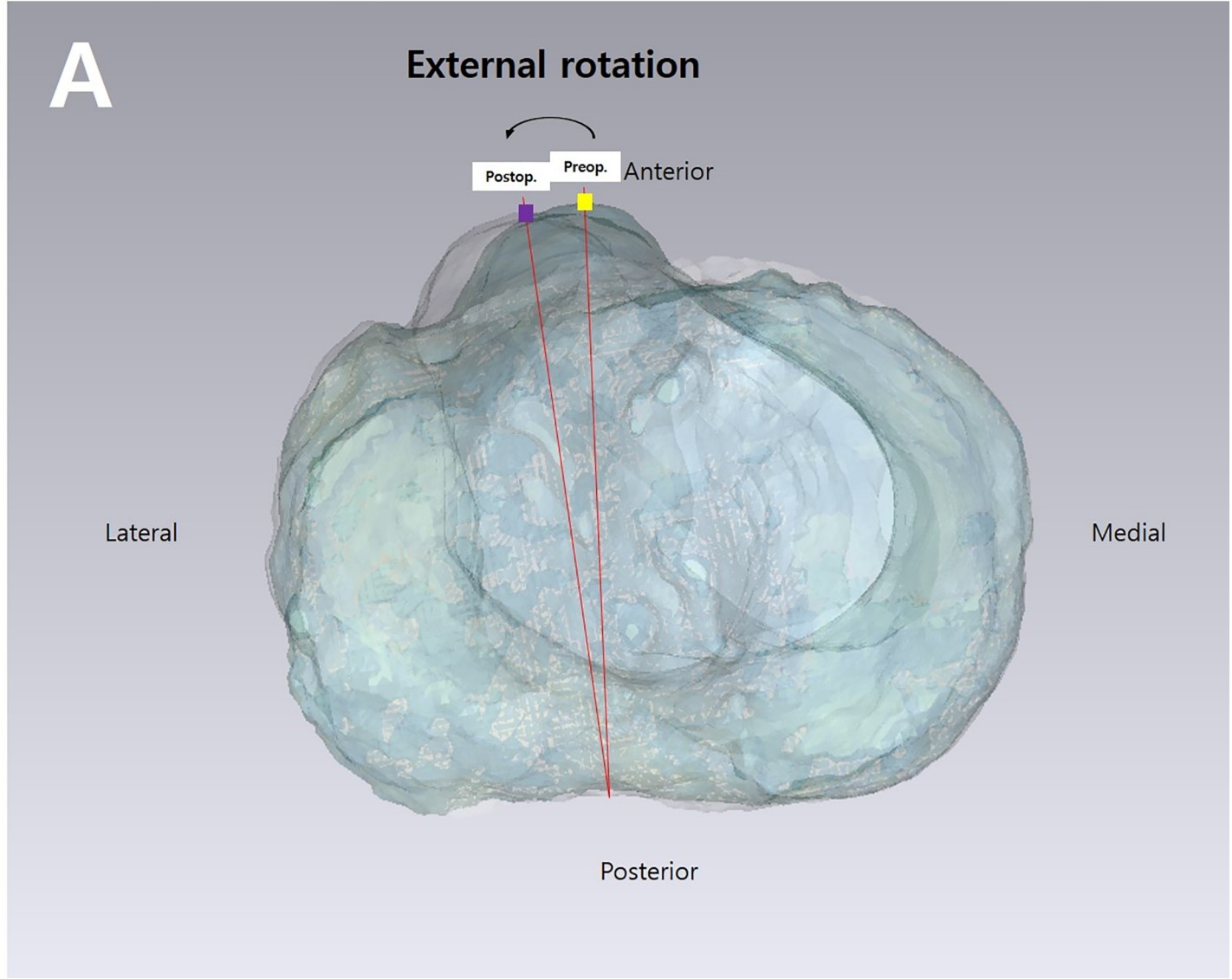

**Fig 3. Measuring the angle of the axial rotational change of the distal tibia after HTO.** Two lines were drawn from the sulcus of the posterior border of the tibial plateau to the marked preoperative and postoperative tibial tuberosity positions overlaid on the tibial plateaus of the preoperative and postoperative three-dimensional model on the axial planes.

preoperative and postoperative tibial tubercles (Fig 3). The captured true axial view of the overlay image was imported into Image J v.1.45s (National Institutes of Health, Bethesda, MD, USA). The angle of the axial rotational change was measured subsequently.

## Statistical analysis

The primary study outcomes were to determine whether distal tibial rotation was associated with opening gap width across all subjects. Using an $\alpha$-level of 0.05 and a power of 0.8, a power analysis was performed to show the statistical significance of the correlation coefficient (0.392) between the distal tibial rotation and the opening gap width, based on the findings of this correlation coefficient in a pilot study involving 6 subjects. Power analysis showed that 38 knees were required to show a statistically significant relationship between distal tibial rotation

and opening gap width. This study finally included 42 knees, indicating adequate power (0.875) to detect a significant correlation between distal tibial rotation and the opening gap width.

All statistical analyses were performed using IBM SPSS Statistics version 20 software (IBM Corp., Armonk, NY, USA). A *P* value < 0.05 was considered statistically significant. All data are presented as means and standard deviations. The intra- and interobserver reliabilities of each measurement were assessed by determining the intraclass correlation coefficient (ICC) and standard error of measurement. The single measured ICC was used to determine the intraobserver reliability of measurements obtained on two occasions by each observer. The average measured ICC was used to evaluate interobserver reliability by comparing the mean of two measurements of each variable.

Demographic characteristics, alignment correction amount, angle of vertical osteotomy direction, biplane osteotomy angle, and medial opening gap width were compared using Student's *t*-tests or Mann-Whitney *U* tests, as appropriate. The rate of lateral hinge fractures was compared using Chi-square test. The correlations between distal tibial rotation and other parameters, such as opening gap amount, angle of vertical osteotomy direction, biplane osteotomy angle, and preoperative varus deformity, were assessed using Pearson correlation analysis. Multiple linear regression analyses were performed to identify the predictor of distal tibial rotation. Receiver operating characteristic analysis was used to determine the cutoff value of the opening gap width that would cause > 3˚ distal tibial rotation.

## Results

The interobserver and intraobserver reliabilities of mechanical axis (MA) measurements on full-length standing radiographs, determinations of the absence or presence of lateral hinge fractures, and distal tibial rotation measurements on the postoperative 3-D model ranged from 0.758 to 0.848, indicating good reliability. The mean preoperative MA was varus 6.7˚ ± 3.4˚ (range, 2.5˚ to 15.7˚), and the mean postoperative MA was valgus 2.9˚ ± 2.4˚ (range, valgus 11.7˚ to varus 2.1˚), making the mean MA correction 9.7˚ ± 4.1˚ (range, 1.9˚ to 19.4˚). Postoperative CT scans showed that 15 of the 42 knees had lateral hinge fractures. All 15 were classified as type I, with none classified as type II or III.

The mean distal tibial rotation was 2.7˚ ± 2.3˚ externally (range, -0.9˚ to 9.9˚). Five of the 42 subjects (12%) showed internal rotation of the distal tibia, and the range of internal rotation of these five cases was less than 1˚ in all cases; the remaining 37 cases showed external rotation. When considering the 3˚ criteria of the acceptable range of distal tibial rotation, 24 cases were less than 3˚ (57%). and the remaining 18 cases (43%) were greater than 3˚. While there was no significant difference in demographics, pre- and postoperative MA, lateral hinge fracture, angle of vertical osteotomy direction, biplane osteotomy angle, or the alignment correction amount between the groups with > 3˚ and ≤ 3˚ distal tibial rotation, the opening gap width was larger in the group with > 3˚ distal tibial rotation than in the group with ≤ 3˚ distal tibial rotation (11.4 mm vs. 9.6 mm, P = 0.027, Table 2).

The preoperative varus (*r* = 0.402, *P* = .028), angle of vertical osteotomy direction angle (r = 0.342, *P* = 0.027) and opening gap width (*r* = 0.294, *P* = .006) were associated with distal tibial rotation on correlation analysis (Table 3).

However, on multiple regression analysis, the opening gap width was the only predictor of distal tibial rotation (*β* = 0.418, *P* = .006, Table 4).

Receiver operating characteristics analysis was performed to determine the opening gap width threshold value predictive of > 3˚ distal tibial rotation. The area under the curve was 0.698 (95% confidence interval, 0.535 to 0.861) for > 3˚ distal tibial rotation. An opening gap

**Table 2. Demographic data, opening gap, and preoperative and postoperative alignment in knees with rotation greater than 3˚ and less than 3˚.**

|  | Dital tibial rotation ≤ 3˚ (n = 24) | Dital tibial rotation > 3˚ (n = 18) | P-value |
|---|---|---|---|
| Age | 57.1 ± 6.9 | 56.2 ± 7.4 | 0.669 |
| BMI | 27.4 ± 3.2 | 27.7 ± 3.4 | 0.791 |
| Opening gap with | 9.6 ± 2.9 | 11.4 ± 2.4 | 0.027 |
| Preoperative MA | 6.0 ± 3.0 | 7.6 ± 2.5 | 0.078 |
| Postoperative MA | -3.4 ± 2.5 | -3.3 ± 2.0 | 0.833 |
| Alignment correction | 9.5 ± 3.9 | 10.9 ± 3.2 | 0.192 |
| Lateral hinge fractures | 9 | 6 | 0.780 |
| Angle of vertical osteotomy direction | 23.1 ± 8.6 | 28.3 ± 7.6 | 0.051 |
| Biplane osteotomy angle | 105.1 ± 9.4 | 107.1 ± 8.4 | 0.468 |

BMI, body mass index; MA, mechanical axis

width of 10 mm was found to be the optimal cutoff value for achieving greater than 3˚ of distal tibial rotation, with a sensitivity of 61% and a specificity of 75% (Fig 4).

## Discussion

The most important finding of this study was to reveal the amount of distal tibial external rotation below the osteotomy gap in open wedge HTO and the positive association between the amount of distal tibial rotation and the opening gap.

Previous in vivo studies by Jang et al.[16] and Hinterwimmer et al.[17] reported that the distal tibia below the osteotomy site rotated internally from 3˚ o 4.5˚, while the current study showed that the distal tibia rotated externally 2.7˚ following open wedge HTO. There are some possible factors that affect the distal tibial rotation. Firstly, once the osteotomy site was spread open, tension could occur around the intact fibular or tibiofibular joint.[7] Unless the fibular or tibiofibular joint was disrupted, tibial rotation would inevitably develop to some extent in order to relieve that tension by opening the osteotomy site gap. The direction of the rotation of the distal tibial fragment could be determined by the tension in the soft tissue envelope of the proximal tibia, such as the hamstring tendon, medial collateral ligament (MCL), or joint capsule. The hamstring tendon acted as an internal rotator of the tibia and the medial collateral ligament to prevent tibial rotation. Therefore, differences in managing these key medial structures of the proximal tibia during the surgery could affect distal tibial rotation following open wedge HTO. The two studies mentioned above by Jang et al.[16] and Hinterwimmer et al.[17]

**Table 3. Correlations between distal tibial rotation and pre- and postoperative alignment, opening gap, and alignment correction before and after opening wedge high tibial osteotomy.**

| | Distal tibial rotation | |
|---|---|---|
| | Correlation coefficient | P-value |
| Preoperative MA | 0.402 | 0.028 |
| Opening gap width | 0.418 | 0.006 |
| Postoperative MA | 0.075 | 0.636 |
| Alignment correction | 0.272 | 0.082 |
| Angle of vertical osteotomy direction | 0.342 | 0.027 |
| Biplane osteotomy angle | 0.199 | 0.207 |

MA, mechanical axis

**Table 4. Multiple linear regression analysis of factors affecting the change of the posterior slope after opening wedge HTO in all included subjects.**

| Dependent Variables | Independent Variables | Non-standardized Coefficients | | Standardized Coefficients | |
|---|---|---|---|---|---|
| | | B | SE | β | P-value |
| Distal tibial rotation | Age | -0.003 | 0.050 | -0.058 | 0.954 |
| | Preoperative MA | 0.091 | 0.247 | 0.114 | 0.715 |
| | Opening gap width | 0.341 | 0.117 | 0.418 | 0.006 |
| | Postoperative MA | 0.265 | 0.245 | 0.250 | 0.286 |
| | Alignment correction | -0.113 | 0.166 | -0.168 | 0.499 |
| | Angle of vertical osteotomy direction | 0.069 | 0.040 | 0.253 | 0.094 |
| | Biplane osteotomy angle | 0.048 | 0.039 | 0.184 | 0.229 |

B, unstandardized coefficients; SE, standard error; β, standardized coefficients; MA, mechanical axis

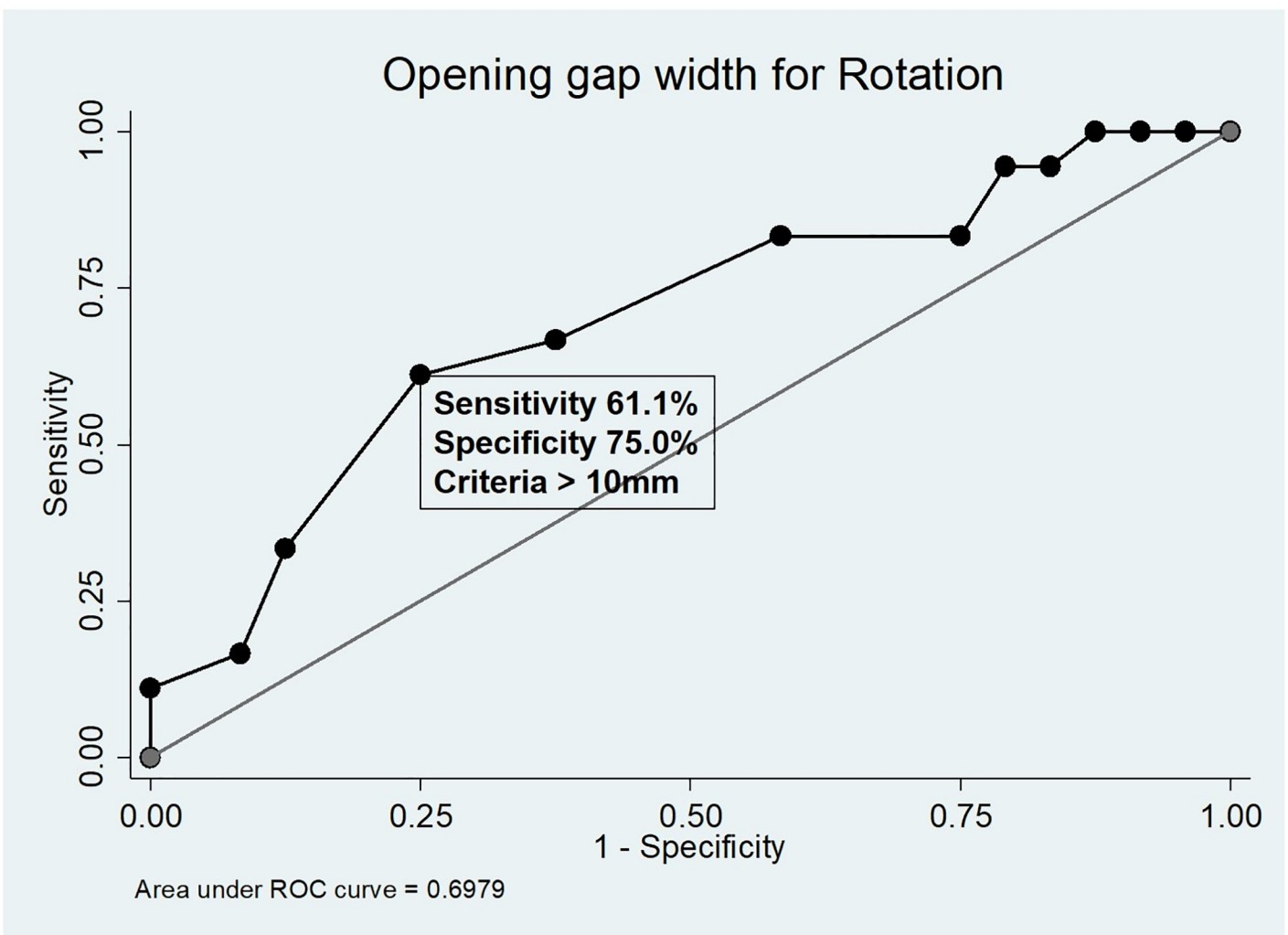

**Fig 4. Receiver operating characteristics analysis.** Receiver operating characteristic curves illustrating the optimal cut-off values for opening gap width predicting > 3° distal tibial external rotation.

showed partial release not only of the hamstring tendon but also of the MCL at the tibial insertion, so the distal part of their insertion could remain at the distal tibia. This remaining hamstring could retract the distal tibia, which was relatively unstable due to partial release of the MCL. However, our technique was different from that used in the above two studies in that the hamstring tendon was detached completely from the tibial insertion site, and in most cases. The difference in hamstring management could explain in part the different direction and amount of distal tibial rotation; the internal rotation was relatively large (4° to 5°) in the previous two studies, and the external rotation was small (2.7°) in our study. Also, the direction of vertical osteotomy in axial plane might be reason for external rotation of distal tibia in the study. According to the current result, the vertical osteotomy was directed from anteromedial to posterolateral in axial plane, which might result in external rotation of distal tibia. Despite statistically insignificance, the group of dital tibial rotation > 3° tended to have larger angle of vertical osteotomy direction, representing more posterolateral direction possibly causing external rotation, than the group of distal tibial rotation ≤ 3° (P = 0.051). In addition, the difference in osteoarthritis severity and the magnitude of posterior capsular release could also influence the direction and amount of distal tibial rotation. A recent cadaveric study[18] that evaluated factors influencing distal tibial rotation in open wedge HTO showed that distal tibial rotation was 1.4° less (more internal) in an advanced osteoarthritic knee than in a mild osteoarthritic knee because of the fact that the posterior capsule of an advanced osteoarthritic knee is stiffer than that of a mild osteoarthritic knee, increasing the risk of insufficient posteromedial capsular release in an advanced osteoarthritic knee.

Interestingly, the results of our study demonstrated that the opening gap width was associated with distal tibial rotation. It is clear that the larger the correction amount, the more freedom is present for tibial rotation to change.[18] Furthermore, a large opening gap could lead to a lateral hinge fracture,[19–21] which could make the distal tibial portion more unstable. Subsequently, this situation could cause the distal tibia to rotate more easily and unpredictably. Therefore, from the viewpoint of the orthopedic surgeon, it is important to determine how much the medial opening gap could increase the risk of tibial rotation. The results of the current study hint at a solution, to a certain degree, by suggesting a 10 mm opening gap as a threshold causing > 3° external rotation of the distal tibia. Therefore, because of the risk of excessive external rotation, the orthopedic surgeon should carefully open the gap slowly and gradually when a gap greater than 10 mm is needed.

The clinical implications of the current study are highlighted in a recent biomechanical study by Yazdi et al.[22] that investigated the effect of tibial torsion on the contact pressures of the knee joint. That study showed that 15° of internal rotation of the distal tibia below the osteotomy site increased the medial compartment contact pressure by 18% compared with neutral rotation of the distal tibia. However, with 15° of external rotation of the distal tibia, medial compartment contact pressure was decreased by 11% compared to that in the neutral position. However, the osteotomy in that study did not reflect the actual condition of open wedge HTO because the osteotomy was performed at the tibia shaft, and fibulectomy was simultaneously performed between the proximal 2/3 and distal 1/3 of the fibula to harvest 2 cm of bone to facilitate tibial rotation. In another more recent biomechanical study, Suero et al.[23] reported that fixation of the distal tibial fragment at 15° of external rotation negated the intended beneficial effect of offloading the medial compartment in open wedge HTO. Taking the results of these two biomechanical studies together, although external rotation of the distal tibia might be less harmful to the biomechanical environment of the knee joint than internal rotation of the distal tibia, it is better to prevent distal tibial rotation even of a small magnitude. In this sense, our results alert orthopedic surgeons to the possibility of external

distal tibial rotation, even as small as 3˚, following open wedge HTO, thus encouraging the orthopedic surgeon to try to avoid causing distal tibial rotation during surgery.

This study has several limitations. One of the most important limitations was that the unpredictable factors involved in osteotomy that might affect the magnitude and direction of distal tibial rotation could not entirely be standardized in all patients. The remaining bone stock from the lateral cortex on the osteotomy plane or from the articular tibial plateau and position of the hinge point or the relationship between the osteotomy direction and the hinge point,[21,24,25] unavoidably differed to a certain degree between subjects because these parameters may have determined the direction of stress within the proximal tibia, which could lead to development of unpredictable fracture line propagation around the osteotomy site, influencing the direction and amount of distal tibial rotation. However, all procedures in our study were performed by one surgeon who tried to standardize the surgical technique parameters, such as hinge point, plate position, and osteotomy direction. This may minimize the bias related to the surgical technique. In addition, we used a cutoff value of 3˚ because it seemed to have clinical relevance, but we did not have proper evidence to support this decision. However, previous studies also suggested that a cutoff value of rotation of 5˚ to 10˚ could alter knee joint kinematics after open wedge HTO,[18] but we wanted a more strict criterion for distal tibial rotation than that used by previous studies. Another limitation was that all subjects included in this study underwent biplane open wedge osteotomy; hence, it is difficult to directly apply the results of our study to uniplane open wedge osteotomy or closed wedge osteotomy.

## Conclusion

The distal tibia below the osteotomy site rotated externally approximately 3˚ following open wedge HTO, and the magnitude of external rotation at distal tibial portion was affected by the opening gap.

## Supporting information

**S1 File. Data sheet of patients' information.**
(XLSX)

## Author Contributions

**Conceptualization:** Jun-Ho Kim.

**Data curation:** Jun-Ho Kim, Hoon-Young Kim, Dae-Hee Lee.

**Formal analysis:** Jun-Ho Kim, Dae-Hee Lee.

**Investigation:** Dae-Hee Lee.

**Methodology:** Jun-Ho Kim, Hoon-Young Kim, Dae-Hee Lee.

**Resources:** Dae-Hee Lee.

**Supervision:** Dae-Hee Lee.

**Validation:** Dae-Hee Lee.

**Writing – original draft:** Jun-Ho Kim.

**Writing – review & editing:** Jun-Ho Kim, Dae-Hee Lee.

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
