## [Decision Letter · Decision Letter 0]

30 Oct 2019

PONE-D-19-17787

Opening gap width influences distal tibial rotation below the osteotomy site following open wedge high tibial osteotomy

PLOS ONE

Dear Dr Lee,

Thank you for submitting your manuscript to PLOS ONE. After careful consideration, we feel that it has merit but does not fully meet PLOS ONE’s publication criteria as it currently stands. Therefore, we invite you to submit a revised version of the manuscript that addresses the points raised during the review process.

We would appreciate receiving your revised manuscript by Dec 14 2019 11:59PM. To enhance the reproducibility of your results, we recommend that if applicable you deposit your laboratory protocols in protocols.io, where a protocol can be assigned its own identifier (DOI) such that it can be cited independently in the future. For instructions see: http://journals.plos.org/plosone/s/submission-guidelines#loc-laboratory-protocols

We look forward to receiving your revised manuscript.

Kind regards,

David Fyhrie

Academic Editor

PLOS ONE

Journal Requirements:

1. Thank you for including your ethics statement; "We received institutional review board approval for the study from the authors institution. "

2. Please provide additional details regarding participant consent. In the ethics statement in the Methods and online submission information, please ensure that you have specified (1) whether consent was informed and (2) what type you obtained (for instance, written or verbal). If your study included minors, state whether you obtained consent from parents or guardians. If the need for consent was waived by the ethics committee, please include this information.

4. Thank you for including your fudning statement; "No. The funders had no role in study design, data collection and analysis, decision to publish, or preparation of the manuscript."

Please provide an amended Funding Statement that declares *all* the funding or sources of support received during this specific study (whether external or internal to your organization) as detailed online in our guide for authors at http://journals.plos.org/plosone/s/submit-now.  

Please state what role the funders took in the study.  If any authors received a salary from any of your funders, please state which authors and which funder. If the funders had no role, please state: "The funders had no role in study design, data collection and analysis, decision to publish, or preparation of the manuscript."

Additional Editor Comments (if provided):

Please pay particular attention to statistical issues raised by reviewer.

Reviewers' comments:

Reviewer's Responses to Questions

**Comments to the Author**

1. Is the manuscript technically sound, and do the data support the conclusions?

Reviewer #1: Partly

2. Has the statistical analysis been performed appropriately and rigorously? 

Reviewer #1: No

3. Have the authors made all data underlying the findings in their manuscript fully available?

Reviewer #1: Yes

4. Is the manuscript presented in an intelligible fashion and written in standard English?

Reviewer #1: Yes

5. Review Comments to the Author

Reviewer #1: Q1. Regarding the hinge, you described that efforts were made to keep the lateral cortex and lateral capsular hinge intact (lines 94 to 95). However, the actual hinge situation was not reported. The rate of hinge fracture is said to be at least over 20% when evaluated CT scans. Fortunately, you have all CT data for making the 3D analysis. Could you examine the presence or absence of the hinge fracture and the type of the fracture according to Takeuchi classification? As you mentioned, large opening gap tend to make unstable hinge fracture. If the rate of hinge fracture was high in the large opening gap group than in the small opening gap group, hinge fracture could be the predictor of the distal tibial rotation.

Q2. You compared the demographic data of the two groups (distal tibial rotation <3 degrees and distal tibial rotation>3 degrees) including opening gap width, preoperative MA, postoperative MA, and alignment correction (Table 2). However, the alignment correction is estimated by the subtraction of the preoperative MA from the postoperative MA. Furthermore, the alignment correction would be calculated by the opening gap and the tibial width. Therefore, these parameters have too many confounding biases each other. On the contrary, there would be more parameters which could have some relationships with tibial rotation; such as the angle between the ascending cut and the transverse cut, the direction of the ascending cut on the axial slice, or lateral hinge fracture, and so on. Reconsider the necessary parameters to be compared in the both groups.

Q3. As you mentioned in the discussion section (lines 243 to 246), hamstring tendon acts as an internal rotator. Therefore how to treat the pes and MCL in OWHTO may affect the tibial rotation. Could you describe more detailed procedure regarding medial soft tissues in the Surgical Technique section? There seems to be some contradictions suggested in the following Q4.

Q4. In your technique, the oblique osteotomy line, i.e. transverse cut, is started from 5 to 6 cm from the joint line. In general, about 4cm is recommended for the starting point.

If you cut from 5 to 6cm from the joint line,

1) The cutting line is far below from the attachment of the hamstring tendons. Therefore the larger amount of the release in the pes would be required. It may create larger external rotation of the tibia.

2) You mentioned that the MCL distal insertion remained intact because the osteotomy was performed just distal to the insertion site of the MCL. However, the superficial MCL attaches 7 to 8 cm from the joint line. Therefore, when applying the 5 to 6cm cutting line, the MCL could be cut during the osteotomy.

6. PLOS authors have the option to publish the peer review history of their article (what does this mean?). If published, this will include your full peer review and any attached files.

Reviewer #1: No

---

## [Author Response · Author response to Decision Letter 0]

19 Dec 2019

Dear Editor, 

Thank you for your kind consideration regarding our paper. 

We revised our manuscript according to the reviewers’ comments. Details of our responses are below and changes in the manuscript are attached with additional file.

We look forward to hearing from you regarding this paper.

---

## [Decision Letter · Decision Letter 1]

6 Jan 2020

Opening gap width influences distal tibial rotation below the osteotomy site following open wedge high tibial osteotomy

PONE-D-19-17787R1

Dear Dr. Lee,

We are pleased to inform you that your manuscript has been judged scientifically suitable for publication and will be formally accepted for publication once it complies with all outstanding technical requirements.

With kind regards,

David Fyhrie

Academic Editor

PLOS ONE

Additional Editor Comments (optional):

Reviewers' comments:

Reviewer's Responses to Questions

**Comments to the Author**

1. If the authors have adequately addressed your comments raised in a previous round of review and you feel that this manuscript is now acceptable for publication, you may indicate that here to bypass the “Comments to the Author” section, enter your conflict of interest statement in the “Confidential to Editor” section, and submit your "Accept" recommendation.

Reviewer #1: All comments have been addressed

2. Is the manuscript technically sound, and do the data support the conclusions?

Reviewer #1: Yes

3. Has the statistical analysis been performed appropriately and rigorously? 

Reviewer #1: Yes

4. Have the authors made all data underlying the findings in their manuscript fully available?

Reviewer #1: Yes

5. Is the manuscript presented in an intelligible fashion and written in standard English?

Reviewer #1: Yes

6. Review Comments to the Author

Reviewer #1: I regret your wrong description on the surgical thchnique. Please check the whole manuscript with your co-authors. However, the suggested points were all corrected.

7. PLOS authors have the option to publish the peer review history of their article (what does this mean?). If published, this will include your full peer review and any attached files.

Reviewer #1: No

---

## [Editor Report · Acceptance letter]

9 Jan 2020

PONE-D-19-17787R1 

Opening gap width influences distal tibial rotation below the osteotomy site following open wedge high tibial osteotomy 

Dear Dr. Lee:

I am pleased to inform you that your manuscript has been deemed suitable for publication in PLOS ONE. Congratulations! Your manuscript is now with our production department. 

With kind regards,

on behalf of

Dr. David Fyhrie 

Academic Editor

PLOS ONE